# Ongoing Gaps in the Hepatitis C Care Cascade during the Direct-Acting Antiviral Era in a Large Retrospective Cohort in Canada: A Population-Based Study

**DOI:** 10.3390/v16030389

**Published:** 2024-03-01

**Authors:** Ana Maria Passos-Castilho, Donald G. Murphy, Karine Blouin, Andrea Benedetti, Dimitra Panagiotoglou, Julie Bruneau, Marina B. Klein, Jeffrey C. Kwong, Beate Sander, Naveed Z. Janjua, Christina Greenaway

**Affiliations:** 1Centre for Clinical Epidemiology, Lady Davis Institute, Jewish General Hospital, Montreal, QC H3T 1E2, Canada; 2Department of Medicine, McGill University, Montreal, QC H3G 2M1, Canada; 3Laboratoire de Santé Publique du Québec, Institut National de Santé Publique du Québec, Sainte-Anne-de-Bellevue, QC H9X 3R5, Canada; donald.murphy@inspq.qc.ca; 4Unité sur les Infections Transmissibles Sexuellement et par le Sang, Institut National de Santé Publique du Québec, Québec, QC H9X 3R5, Canada; karine.blouin@inspq.qc.ca; 5Department of Epidemiology, Biostatistics and Occupational Health, School of Population and Global Health, McGill University, Montreal, QC H3A 1G1, Canada; andrea.benedetti@mcgill.ca (A.B.); dimitra.panagiotoglou@mcgill.ca (D.P.); 6CHUM Research Centre, Centre Hospitalier de l’Université de Montréal, Montreal, QC H2X 0A9, Canada; julie.bruneau.umontreal@gmail.com; 7Research Institute of the McGill University Health Centre, Montreal, QC H3H 2R9, Canada; marina.klein@mcgill.ca; 8Dalla Lana School of Public Health, University of Toronto, Toronto, ON M5T 3M7, Canada; jeff.kwong@utoronto.ca; 9ICES, Toronto, ON M4N 3M5, Canada; beate.sander@uhnresearch.ca; 10Public Health Ontario, Toronto, ON M5G 1M1, Canada; 11Department of Family and Community Medicine, University of Toronto, Toronto, ON M5G 1V7, Canada; 12Toronto General Hospital Research Institute, University Health Network, Toronto, ON M5G 2C4, Canada; 13Institute of Health Policy, Management and Evaluation (IHPME), University of Toronto, Toronto, ON M5T 3M6, Canada; 14School of Population and Public Health, University of British Columbia, Vancouver, BC V6T 1Z3, Canada; naveed.janjua@bccdc.ca; 15Division of Infectious Diseases, Jewish General Hospital, Montreal, QC H3T 1E2, Canada

**Keywords:** hepatitis C, care cascade, birth cohort, people who inject drugs, immigrants

## Abstract

To achieve hepatitis C virus (HCV) elimination, high uptake along the care cascade steps for all will be necessary. We mapped engagement with the care cascade overall and among priority groups in the post-direct-acting antivirals (DAAs) period and assessed if this changed relative to pre-DAAs. We created a population-based cohort of all reported HCV diagnoses in Quebec (1990–2018) and constructed the care cascade [antibody diagnosed, RNA tested, RNA positive, genotyped, treated, sustained virologic response (SVR)] in 2013 and 2018. Characteristics associated with RNA testing and treatment initiation were investigated using marginal logistic models via generalized estimating equations. Of the 31,439 individuals HCV-diagnosed in Quebec since 1990 and alive as of 2018, there was significant progress in engagement with the care cascade post- vs. pre-DAAs; 86% vs. 77% were RNA-tested, and 64% vs. 40% initiated treatment. As of 2018, a higher risk of not being RNA-tested or treated was observed among individuals born <1945 vs. >1965 [hazard ratio (HR); 95% CI; 1.35 (1.16–1.57)], those with material and social deprivation [1.21 (1.06–1.38)], and those with alcohol use disorder [1.21 (1.08–1.360]. Overall, non-immigrants had lower rates of RNA testing [0.76 (0.67–0.85)] and treatment initiation [0.63 (0.57–0.70)] than immigrants. As of 2018, PWID had a lower risk of not being RNA tested [0.67 (0.61–0.85)] but a similar risk of not being treated, compared to non-PWID. Engagement in the HCV care cascade have improved in the post-DAA era, but inequities remain. Vulnerable subgroups, including certain older immigrants, were less likely to have received RNA testing or treatment as of 2018 and would benefit from focused interventions to strengthen these steps.

## 1. Introduction

The elimination of hepatitis C virus (HCV) infection as a public health threat is possible given the availability of curative, well-tolerated, oral direct-acting antivirals (DAAs). To achieve this goal, a high uptake in all steps of the HCV care cascade (diagnosis, linkage to care, treatment, cure) for all groups at risk will be necessary [1]. The populations that account for the greatest number of HCV cases in Canada [people who inject drugs (PWID), immigrants born in countries where HCV is common, and individuals born during 1945–1965] are diverse, and each have unique challenges and barriers to accessing and engaging in healthcare [2]. Knowledge of the specific gaps and predictors of loss in each step of the HCV care cascade for each priority population is needed to design effective interventions for each group [2].

Canada signed onto the World Health Organization’s (WHO) goal to eliminate hepatitis C as a public health threat by 2030. With less than 10 years left, real-world population-based data across the country are urgently needed to measure gaps and prioritize health service delivery and policy. The HCV care cascade has been reported in Canada in the provinces of British Columbia (BC) and Ontario, with overall improvements in the post-DAA era [3,4,5]. However, there are limited data stratified by PWID or immigration status, not only from Canada but globally [6,7,8,9,10,11]. Our study in Quebec, the second most populous province in Canada, reports the HCV care cascade overall and among PWIDS and immigrants in Canada in the post-DAA era. The data on immigrants disaggregated by region of origin shown in this study can inform policies not only in Canada but also in other high-income countries that receive large numbers of immigrants born in countries with higher anti-HCV prevalence [12]. We characterized the population-level HCV care cascade in a large retrospective cohort in Canada before and after the introduction of DAA treatment overall and among key priority groups. We investigated the predictors of progression through the key cascade steps to identify which subgroups were still not RNA-tested or treated as of 2018 and assess if this changed from 2013. 

## 2. Materials and Methods

### 2.1. Study Design, Cohort Construction, and Variable Definitions

We conducted a population-based, retrospective cohort study of all reported hepatitis C diagnoses in Quebec, Canada, between 1 January 1990 and 31 December 2018 in the reportable disease database (MADO) or the Quebec Public Health Laboratory (LSPQ) database, linked to several provincial administrative datasets, including immigrant landing data, medical visits, hospitalizations, cancers, prescription drugs, and deaths. All reported HCV cases (based on positive antibody or RNA tests) were linked deterministically using a unique health care number to the provincial health insurance registry (FIPA) and all healthcare utilization and cancer databases (1987–2018): (1) physician billings (RAMQ-PB), (2) pharmaceutical services (RAMQ-Pharm), (3) hospitalizations (MED-ECHO), and (4) Quebec Cancer Registry (RQC). Deaths were ascertained through linkage with the Death Registry (ISQ) (1990–2015) and RAMQ-FIPA (2016–2018) databases. All immigrants present in the Quebec Permanent Landed Immigrant Database (MIFI) from 1 January 1980 to the end of the study period were linked deterministically to FIPA through a unique immigration number.

Sociodemographic characteristics at the time of HCV diagnosis included age, sex, and the public health region of residence. Urban was defined as living in Greater Montreal and Quebec City, and rural was all other regions. The Material and Social Deprivation Index (MSDI) was used for neighbourhood material and social deprivation [13]. Individuals were classified into three birth cohorts: before 1945, 1945–1965, and after 1965. Immigrants were defined as people born outside of Canada with permission to live permanently in Canada. The country of origin was grouped into regions according to the World Bank classification [14,15]. PWID (past or present) status [16], human immunodeficiency virus (HIV) co-infection [17], decompensated cirrhosis (DC), and hepatocellular carcinoma (HCC) [3] were identified using published algorithms based on the international classification of diseases (ICD) 9/10 codes and the presence of disease-specific medications (Appendix A) [16,18,19,20,21,22]. Data on HCV-specific antiviral dispensations were obtained from the RAMQ-Pharm database (Appendix A) [23,24,25]. Hepatitis B virus (HBV) co-infection was obtained from laboratory-confirmed diagnoses in the MADO database. We used a three-year lookback period for identifying conditions at the time of HCV diagnosis (baseline). Second-generation, interferon-free DAA therapy was first introduced in Canada in 2014, with unrestricted access in Quebec as of March 2016 [26,27,28]. We defined the pre-DAA period as 1990–2013 and the post-DAA period as 2014–2018, as most (64%) of the treatment initiations in 2014 were DAA medications.

This study was reviewed and approved by the Quebec Information Access Commission (1018218-S) and the research ethics committee of the Integrated Health and Social Services University Network for West–Central Montreal, Quebec (MP-05-2019-1257).

### 2.2. Care Cascade

The HCV care cascades included all individuals diagnosed with HCV since 1990 and alive as of 31 December 2013 (pre-DAA) or 2018 (post-DAA), the years for which care cascades were constructed. The steps along the care cascade included individuals who met the following conditions at any time from HCV diagnosis (1990–2018) to December 31 of the year of the cascade: (1) HCV diagnosed included all confirmed HCV diagnoses (any confirmed positive HCV antibody or a positive HCV RNA); (2) RNA-tested was defined as the presence of the first HCV RNA test in the database after diagnosis; (3) RNA-positive included individuals with a positive HCV RNA test; (4) genotyped is HCV genotype testing performed among RNA-positive individuals, a good proxy for engagement in HCV care and treatment; (5) initiated antiviral treatment was defined as the presence of the first dispensation of any HCV-specific antiviral medications; and (6) achieved a sustained viral response (SVR) was defined as a negative RNA test at least 10 weeks after last antiviral dispensation [3,4]. The backfilling of the care cascade steps is described in Appendix A. The proportion of people at a given step (#1–6) was calculated relative to the number of people in the previous step, except for the proportion of those with an SVR, for which the denominator was the number of individuals assessed for SVR after treatment initiation. Care cascades were estimated overall, for each priority population, and by underlying medical comorbidities.

### 2.3. Statistical Analysis

We performed multiple imputation by chained equations with 20 iterations to address missing data, mostly due to the 11% of unlinked HCV diagnoses (Appendix A) that were assumed to be missing at random [29,30,31]. All statistical analyses were performed on the dataset after multiple imputation, and standard methods were used to pool estimates and standard errors [30,31].

The cumulative failure probability over time from HCV antibody diagnosis to RNA testing and from RNA positivity to treatment initiation was assessed using Kaplan–Meier (KM) curves overall and stratified by priority population. We estimated marginal logistic models via generalized estimating equations (GEEs) to identify characteristics associated with not having been RNA-tested or not having initiated treatment as of 2018 and tested if this changed from 2013 to 2018. Separate models with RNA testing or treatment initiation as the outcome were constructed using the longitudinal data of individuals that were alive as of 2013 and/or 2018, with an exchangeable correlation structure. Initially, all models were tested with interaction terms between the cascade calendar year (2013 or 2018) and each variable. The final models only included interaction terms that were significant at *p* < 0.050. Specific 2013 and 2018 adjusted odds ratios (aORs) for variables with significant interactions, or marginal aORs, are shown. The models were adjusted for baseline confounders selected a priori: sex, birth cohort, PWID status, immigrant status, MSDI, region of residence, year of HCV diagnosis, Elixhauser comorbidity index (any comorbidity/none; modified to exclude comorbidities added as separate variables in the model), and comorbidities (diabetes, liver disease, HIV, alcohol and drug use disorder, major mental health illness diagnosis, and HBV coinfection). In the immigrant-specific models, additional baseline covariates included region of birth, immigration category, and years from admission to HCV diagnosis.

In the sensitivity analysis, adjusted hazard ratios (aHRs) of factors associated with time to HCV diagnosis and HCV treatment initiation were estimated using Fine–Gray subdistribution competing risk hazard regression models with all-cause mortality as the competing event, adjusting for the same covariates as the GEE models [32]. Comorbidities were included as time-varying covariates. Individuals who were RNA-negative or who cleared the infection spontaneously were excluded from the time-to-treatment initiation analysis. The proportional hazard assumption was verified with log–log survival curves and Schoenfeld residual plots of exposure variables over time. All analyses were performed using SAS^®^ 9.4 (SAS Institute Inc., Cary, NC, USA).

## 3. Results

Between 1990 and 2018, 42,514 individuals with an HCV diagnosis were identified, of whom 4565 (11%) did not link to the FIPA database (Appendix A). All the results are reported after multiple imputation. The detailed characteristics of the full cohort, imputed cohort, and individuals with unlinked diagnoses are shown in Appendix A. Of all the HCV-diagnosed individuals, 31,439 (74%) were alive by the end of 2018, among whom the mean age was 41.5 years (SD 12.9), and 66% were male. A total of 55% (n = 17,220) were born during 1945–1965, 45% (n = 14,008) were PWID, and 15% (n = 4862) were immigrants. There was a significant overlap between the priority populations. PWID made up 39% of the 1945–1965 birth cohort, but only 12% of immigrants. Half (50%) of all the immigrants were born during 1945–1965 (Appendix A).

There was substantial improvement in the care cascade from 2013 to 2018 (Figure 1). Of note, 86% vs. 77% of the individuals with an HCV diagnosis received RNA testing, and 70% vs. 45% of the RNA-positive individuals initiated treatment in 2018 vs. 2013, respectively. Table 1 shows the 2018 care cascade stratified by key priority groups, sociodemographic characteristics, and comorbidities. Individuals in the 1945–1965 birth cohort showed better progression through the 2018 care cascade compared to the other birth cohorts (Figure 2A). Among all the HCV-diagnosed PWID, 88% received RNA testing (Figure 2B). The treatment uptake among the genotyped PWID was 68%, with 85% achieving an SVR. Of all the immigrants with an HCV diagnosis, 87% received RNA testing, and 76% of the participants who were genotyped initiated treatment, with 92% achieving an SVR (Figure 2C). The proportion of immigrants who received RNA testing varied greatly by birth cohort, from only 76% of the immigrants born before 1945 to 89% among the immigrants born after 1965 (Appendix A).

### 3.1. Gaps in RNA Diagnosis and Treatment Initiation as of 2018 and Progress between Pre- (2013) and Post-DAA (2018) Periods

The median time to RNA testing after HCV diagnosis as of 2018 was 1.8 years [interquartile range (IQR): 0.3–6.2] overall and 0.2 years (IQR: 0.1–0.5) among the individuals diagnosed in the post-DAA period. The probability of receiving RNA testing after HCV diagnosis among all the HCV-diagnosed individuals was higher and occurred in a shorter period in 2018 compared to 2013. In 2013, it took 17 years to achieve an 80% probability of RNA testing, whereas this was achieved in 11 years in 2018 (Figure 3, panel A). The median time to treatment initiation after an initial positive RNA test up to 2018 was 2.4 years (IQR 0.7–6.6) overall and 0.6 years (IQR 0.2–1.3) among the individuals diagnosed in the post-DAA period. The probability of treatment initiation after a positive RNA test increased over calendar time (Figure 3, panel B). In 2013, it took 22 years to achieve a 50% probability of treatment initiation after a positive RNA test, whereas in 2018, this occurred within 7 years.

The risk of not receiving RNA testing was lower in 2018 (aORs, 95% CI: 0.67, 0.61–0.74) compared with 2013 (0.83, 0.76–0.90) among the PWID relative to the non-PWID patients. Overall, immigrants were at a 24% lower risk (0.76, 0.67–0.85, *p* < 0.001) of not receiving RNA testing as of 2018 relative to non-immigrants. In the sensitivity analysis, including an interaction term between immigrant status and birth cohort by cascade year, older immigrants (born < 1945) did not have this advantage and had a similar risk of not receiving RNA testing as non-immigrants (2013: 1.18, 0.90–1.55; 2018: 1.31, 0.99–1.74). In a separate model for 2018 only, immigrants born prior to 1945 had a 50% increased risk (1.50, 1.15–1.96) of not receiving RNA testing. Other predictors of not receiving RNA testing remained similar from 2013 to 2018; material deprivation, older age (born < 1945), rural residence, and alcohol use disorder (Table 2).

The risk of not receiving treatment as of 2018 decreased significantly for PWID (2013: 1.30, 1.20–1.41; 2018: 1.03, 0.96–1.11) and individuals with HIV coinfection (2013: 0.93, 0.77–1.11; 2018: 0.38, 0.31–0.46) compared to 2013. The 1945–1965 birth cohort, compared to the youngest (0.86, 0.82–0.91) and immigrants (0.63, 0.57–0.70), had a lower risk of not receiving treatment. The risk among immigrants varied significantly, however, and those from Latin America and the Caribbean were approximately 50% more likely to not have initiated treatment as of 2018 compared to those from high income countries in Europe, the United States, Australia, and New Zealand (Appendix A). As of 2018, predictors of not having initiated treatment were older age (born before 1945), alcohol use disorder, material deprivation, and diabetes (Table 2).

### 3.2. Sensitivity Analysis: Factors Associated with Time to RNA Testing and Treatment Initiation in 2018

There was no violation of the proportional hazards assumption in the sensitivity analysis of factors associated with time to RNA testing and treatment initiation in 2018, except for PWID. An interaction term of PWID with time was added to the model to address this violation and to allow the effect of PWID on the outcome to vary over time. RNA testing was lower for the 1945–1965 (aHRs, 95% CI: 0.95, 0.93–0.97) and the <1945 (0.68, 0.64–0.72) birth cohorts compared to the youngest (born > 1965) (Appendix A). RNA testing has changed over time among PWID. It was lower among PWID in the first year after HCV diagnosis (0.88, 0.79–0.97), similar until 7 years post-diagnosis, and then was higher after 7 years (1.38, 1.04–1.84) relative to non-PWID persons (Appendix A). Immigrants had increased RNA testing (1.12, 1.09–1.15) compared to non-immigrants (Appendix A), which was observed across all regions of origin except for immigrants from Latin America and the Caribbean (0.95, 0.86–1.05) (Appendix A). In immigrant-specific models, those from Latin America and the Caribbean had lower RNA testing rates (0.86, 0.74–1.00) compared to immigrants from high-income countries in Europe, the United States, Australia, and New Zealand (Appendix A).

The adjusted hazard of treatment initiation (Appendix A) was 21% lower in the 1945–1965 birth cohort (aHRs, 95% CI: 0.79, 0.77–0.82) and 71% lower in the <1945 (0.29, 0.27–0.32) birth cohort, compared to the >1965 cohort. PWID had lower treatment initiation than non-PWID patients until seven years post-RNA-positive testing (mean aHRs, 95% CI: 0.77, 0.69–0.86) (Appendix A). Immigrants overall and from all regions of origin had higher treatment initiation (aHRs, 95% CI: 1.42, 1.35–1.48) compared to non-immigrants (Appendix A). In immigrant-specific models, however, those from Latin America and the Caribbean had a 16% decreased hazard of treatment initiation (0.82, 0.69–0.98) compared to immigrants from high-income countries in Europe, the United States, Australia, and New Zealand (Appendix A).

## 4. Discussion

There was substantial improvement in engagement with the HCV care cascade after the introduction of DAA therapy, but important gaps remained: 14% of all HCV-antibody-diagnosed individuals had not received RNA testing, and 36% of all RNA-positive individuals had not initiated treatment as of 2018. Older individuals overall and PWID for the first few years after diagnosis were less likely to be RNA-tested or treated compared to the youngest cohort and non-PWID individuals, respectively. Immigrants showed better progression through all steps of the care cascade compared to non-immigrants, with the exception of older immigrants and those from Latin America and the Caribbean. As of 2018, older age (birth cohort < 1945), rural residence, material and social deprivation, and alcohol use disorder remained significant predictors of not receiving RNA testing and treatment.

We observed substantial improvement across all steps in the care cascade from 2013 to 2018, similar to what has been reported in other parts of Canada [3,5,33]. Progression through several steps of the care cascade in our study was significantly better than what has been described in other countries, including the United States, the United Kingdom, France, and Australia [34,35,36,37,38]. A systematic review found that the proportion of individuals with chronic HCV infection (CHC) who received DAA treatment among the general population was higher than in our study in Iceland (95%), Egypt (92%), and Georgia (79%) but lower in the Netherlands (52%), United States (29%), Norway (18%), Sweden (8%), and Denmark (5%) [39].

In our study, 62% of HCV-viremic PWID initiated treatment, which is higher than reported among PWID in Georgia (50%), Australia (37%), the United States (13%), and BC, Canada (40%) [39]. Our analysis suggests that the increased uptake in RNA testing and treatment as of 2018 compared to 2013 particularly benefitted PWID, who as of 2018 were not at a higher risk of not receiving RNA testing and treatment relative to non-PWID, but this group will need to continue to be supported. In the time-to-event sensitivity analysis, PWID were less likely to be RNA-tested in the first year after an HCV diagnosis and less likely to initiate treatment for up to seven years after the initial RNA-positive test. The timely diagnosis and treatment of PWID are essential and can prevent viral sequelae and premature acquisition risk-related and liver-related deaths [40], as well as onward transmission, especially when combined with opioid agonist therapy and high-coverage needle-and-syringe programs [41,42].

There are limited studies describing the HCV care cascade among immigrants, especially after the introduction of DAAs. The immigrants in our study demonstrated higher uptake across all steps of the HCV care cascade compared to the non-immigrants, which is similar to immigrants in the Ontario HCV Care Cascade [5]. A study in France described the care cascade among vulnerable undocumented immigrants and refugees (2007–2017), with similar RNA testing (86% vs. 87%) but lower treatment initiation (57% vs. 71%) among RNA-positive individuals compared to our study [10]. The immigrants in our study were more likely to be RNA-tested and to initiate treatment compared to the non-immigrants, which is similar to other Canadian studies [5,43,44,45]. Disparities remained among the immigrant subgroups in our study, however, as those born prior to 1945 were less likely to receive RNA testing relative to the younger immigrants, and immigrants from Latin America and the Caribbean were less likely to receive RNA testing and treatment compared to immigrants from high-income regions. Long delays in diagnosis and treatment resulting from a lack of systematic screening after arrival in Canada [46], coupled with unique barriers to accessing and navigating healthcare after immigration, make this group a particularly vulnerable and at-risk group for end-stage liver disease (ESLD). Studies in the pre-DAA era demonstrated a higher risk of developing HCC and dying in hospital due to liver-related complications among immigrants compared to those born in Canada [47,48]. A key knowledge gap is the proportion of undiagnosed HCV infection and late HCV diagnosis among immigrants, which may be contributing to the growing burden of liver disease in Canada and other high-income countries. 

Treatment initiation among individuals with HCC and HIV coinfection improved between 2013 and 2018, likely due to the lifting of restrictions on DAA treatment for those with advanced liver fibrosis [26,27,28] and the better tolerability of DAA therapy in combination with antiretroviral therapy [49]. People residing in rural areas were less likely to receive RNA testing in 2018, suggesting a need for additional strategies, such as using telemedicine [50] and/or outreach screening/RNA testing programs in these areas. Individuals born during 1945–1965 were not at a higher risk of not receiving RNA testing or treatment in 2018 in our main analysis. In the time-to-event sensitivity analysis, however, they presented a 5% lower uptake of RNA testing and a 21% lower rate of treatment initiation, respectively, compared to the youngest cohort, probably due to delayed RNA testing and treatment or survival bias.

Finally, our analysis suggests that disparities in RNA testing and treatment remain even after five years of DAA availability among the oldest birth cohort (<1945), those with material and social deprivation, and those with alcohol use disorder. The timely diagnosis and treatment of older individuals is necessary, as this group has high rates of liver-related complications (i.e., DC and HCC), which, in combination with underlying comorbidities such as HIV, obesity, and alcohol use disorder, have become a major cause of death [51,52,53]. Targeted approaches to promoting equitable progress through the care cascade are urgently needed among socially deprived groups and could include community-based tailored interventions, such as partnering with community organizations and local health clinics. These models will need to be embedded into alcohol use services, which have been shown to be successful in Australia [54]. 

The major strength of this study is that it follows a population-based sample over 28 years. Immigrants were identified through linkage with the landed immigrant database, using data from 1980 onwards, and data disaggregated by birth cohort and region of birth are provided. PWID were identified using a validated algorithm. All notified HCV cases in the province were laboratory-confirmed, and there was a high deterministic linkage rate between the databases. Missing data were imputed using robust methods. This study has limitations, including underestimating HCV diagnoses and possible selection bias due to detection in a passive surveillance system, which could be differential as testing may be higher among PWID but lower among immigrants and others with symptomatic liver disease. While it has been estimated that approximately 25% of Canadians living with past or current HCV infection in Canada remain unaware of their status [55], this proportion is expected to be much higher among the 1945–1965 cohort and immigrants. Despite being validated in administrative databases, the use of published diagnostic code algorithms for identifying HIV diagnoses and PWID may have resulted in an underestimation of these conditions. There is a misclassification of immigrant status for immigrants who arrived in the province before 1980 (an estimated 15% of all immigrants living in Quebec) [56] and those who resided in another province before arriving in Quebec. Despite these limitations, these data provide valuable insights about gaps in care and engagement, with which we can assess important HCV care cascade data that can inform micro-elimination strategies and health service needs in Canada and other low-HCV-prevalence settings.

## 5. Conclusions

Engagement in the HCV care cascade increased overall and among key priority groups (PWID, 1945–1965 birth cohort, immigrants) in Quebec in the five years after the introduction of DAA therapy, but inequities remain. RNA testing and treatment were carried out late among PWID and the 1945–1965 birth cohorts, which can result in increased transmission and a high burden of ESLD and premature death. Older individuals (born < 1945), those living with material and social deprivation, and those who have alcohol use disorder were “left behind” in the care cascade, as were older immigrants and those from Latin America and the Caribbean. These groups should be targeted for RNA testing and treatment initiation, as this will be key to achieving HCV elimination. This study may inform elimination efforts in Quebec and other Canadian provinces and is also relevant to other high-income, low-HCV-prevalence settings with similar key HCV priority groups. 

## Figures and Tables

**Figure 1 viruses-16-00389-f001:**
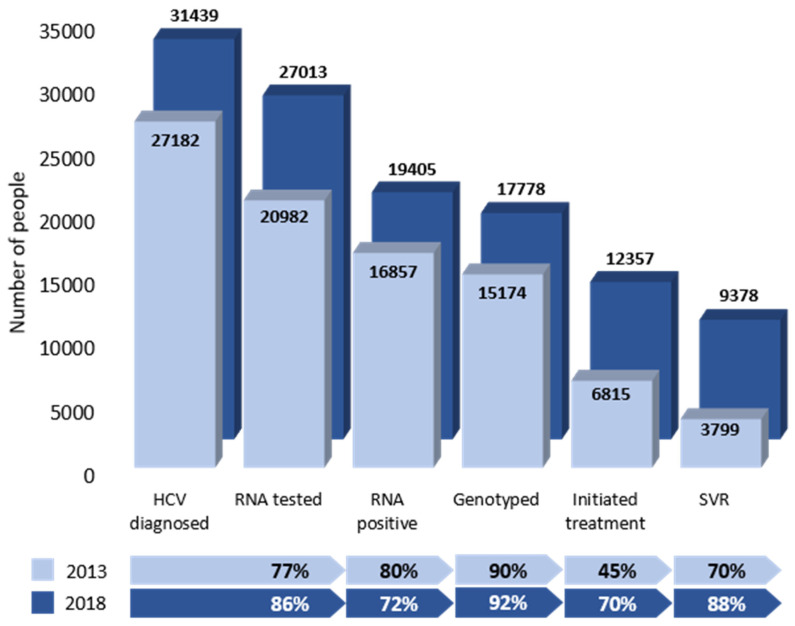
Overall cascade of care of individuals diagnosed with HCV since 1990 and living in Quebec, Canada, as of 2013 compared to 2018 (percentages on the bottom are the proportion of individuals that progressed to the next stage).

**Figure 2 viruses-16-00389-f002:**
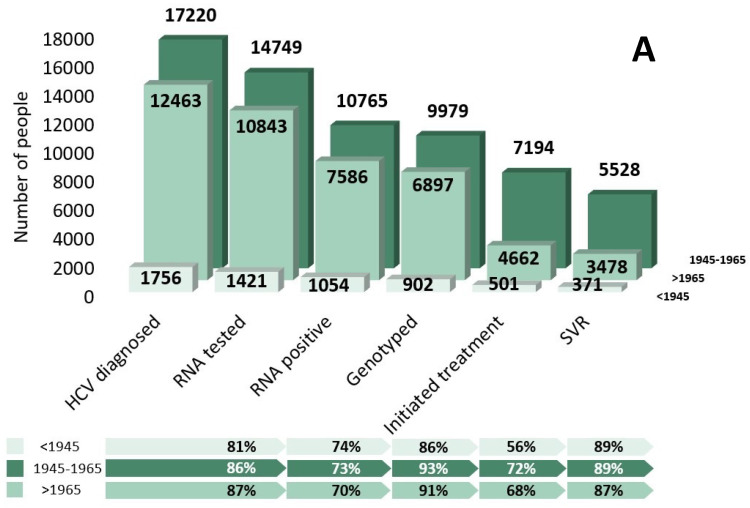
Cascade of care of individuals diagnosed with HCV since 1990 and living in Quebec, Canada, in 2018, stratified by (**A**) birth cohort, (**B**) people who use drugs status, and (**C**) immigrant status (percentages on the bottom are the proportion of individuals that progressed to the next stage).

**Figure 3 viruses-16-00389-f003:**
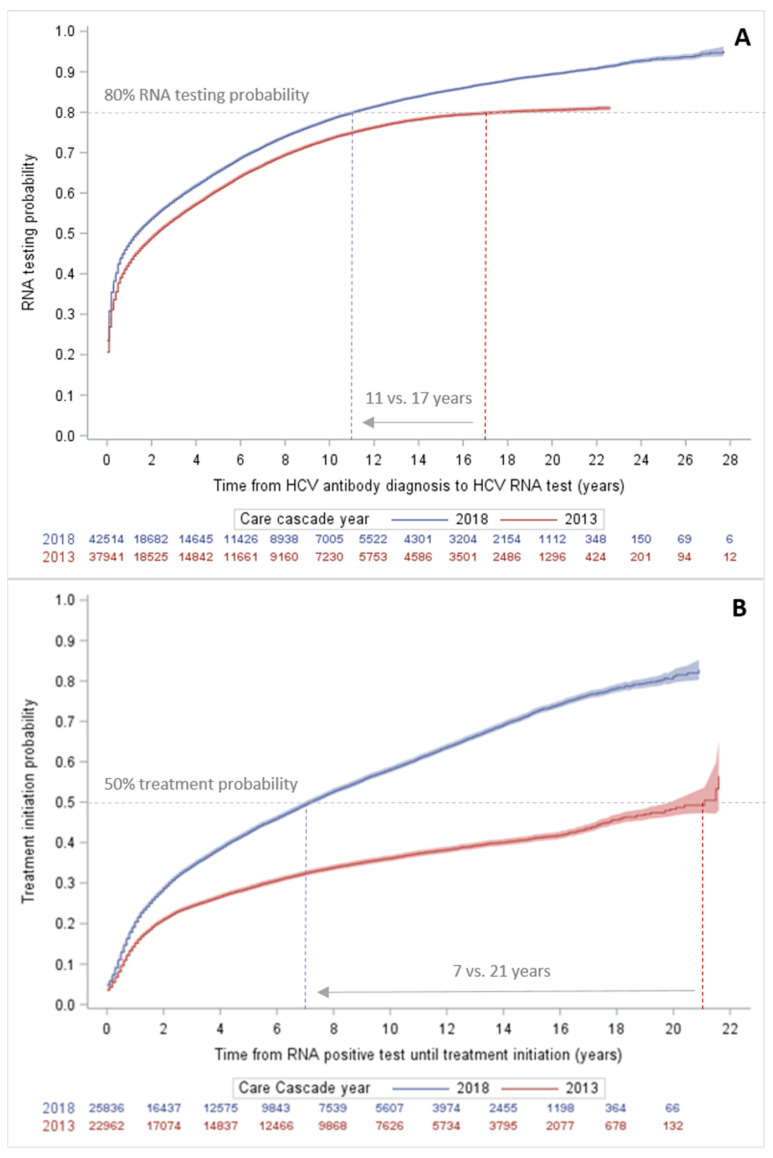
Kaplan–Meyer cumulative failure probability curves with 95% confidence intervals for time from (**A**) HCV diagnosis to RNA testing and (**B**) HCV RNA positive test to treatment initiation; cumulative Quebec HCV care cascades of individuals diagnosed since 1990 and living with HCV as of 2013 (red) and 2018 (blue).

**Table 1 viruses-16-00389-t001:** Overall cascade of care of individuals diagnosed with HCV in 1990–2018 and living in Quebec, Canada, as of 2018, stratified by care cascade stages, with row percentages as a proportion of the previous column.

	HCV-Diagnosed	RNA-Tested	RNA-Positive	Genotyped	Initiated Treatment	Assessed for SVR	SVR
Total	31,439 (100.0)	27,013 (85.9)	19,405 (71.8)	17,778 (91.6)	12,357 (69.5)	10,638 (86.1)	9378 (88.2)
Birth cohort							
<1945	1756 (100.0)	1421 (80.9)	1054 (74.2)	902 (85.6)	501 (55.5)	417 (83.2)	371 (89.0)
1945–1965	17,220 (100.0)	14,749 (85.7)	10,765 (73.0)	9979 (92.7)	7194 (72.1)	6206 (86.3)	5528 (89.1)
>1965	12,463 (100.0)	10,843 (87.0)	7586 (70.0)	6897 (90.9)	4662 (67.6)	4015 (86.1)	3478 (86.6)
Sex							
Female	10,700 (100.0)	9184 (85.8)	6241 (68.0)	5676 (90.9)	3906 (68.8)	3365 (86.1)	3013 (89.5)
Male	20,739 (100.0)	17,829 (86.0)	13,164 (73.8)	12,102 (91.9)	8451 (69.8)	7273 (86.1)	6365 (87.5)
Region of residence							
Urban	20,775 (100.0)	18,036 (86.8)	13,037 (72.3)	11,900 (91.3)	8429 (70.8)	7238 (85.9)	6411 (88.6)
Rural	10,664 (100.0)	8977 (84.2)	6368 (70.9)	5878 (92.3)	3928 (66.8)	3400 (86.6)	2967 (87.3)
Social deprivation quintile							
1 (Least deprived)	3176 (100.0)	2719 (85.6)	1994 (73.3)	1824 (91.5)	1304 (71.5)	1150 (88.2)	1010 (87.8)
2	3932 (100.0)	3399 (86.4)	2457 (72.3)	2260 (92.0)	1602 (70.9)	1393 (87.0)	1220 (87.6)
3	4938 (100.0)	4238 (85.8)	3028 (71.4)	2771 (91.5)	1973 (71.2)	1707 (86.5)	1492 (87.4)
4	7524 (100.0)	6443 (85.6)	4687 (72.7)	4300 (91.7)	2974 (69.2)	2559 (86.0)	2274 (88.9)
5 (Most deprived)	11,869 (100.0)	10,212 (86.0)	7238 (70.9)	6624 (91.5)	4504 (68.0)	3829 (85.0)	3381 (88.3)
Material deprivation quintile							
1 (Least deprived)	4483 (100.0)	3917 (87.4)	2814 (71.8)	2595 (92.2)	1894 (73.0)	1711 (90.3)	1544 (90.2)
2	4801 (100.0)	4122 (85.9)	2989 (72.5)	2776 (92.9)	2003 (72.2)	1723 (86.0)	1541 (89.4)
3	5680 (100.0)	4894 (86.2)	3543 (72.4)	3242 (91.5)	2278 (70.3)	1970 (86.5)	1742 (88.4)
4	6939 (100.0)	5991 (86.3)	4325 (72.2)	3985 (92.1)	2744 (68.9)	2334 (85.1)	2017 (86.4)
5 (Most deprived)	9536 (100.0)	8090 (84.8)	5734 (70.9)	5179 (90.3)	3438 (66.4)	2900 (84.4)	2534 (87.4)
Immigrant status							
Immigrant	4862 (100.0)	4220 (86.8)	3377 (80.0)	3142 (93.0)	2388 (76.0)	1996 (83.6)	1844 (92.4)
Non-immigrant	26,577 (100.0)	22,792 (85.8)	16,028 (70.3)	14,636 (91.3)	9969 (68.1)	8642 (86.7)	7534 (87.2)
People who inject drugs							
Yes	14008 (100.0)	12355 (88.2)	8898 (72.0)	8184 (92.0)	5530 (67.6)	4658 (84.2)	3957 (85.0)
No	17431 (100.0)	14658 (84.1)	10507 (71.7)	9594 (91.3)	6827 (71.2)	5980 (87.6)	5421 (90.7)
Elixhauser comorbidity index (modified) ^a^							
Yes	2115 (100.0)	1828 (86.4)	1289 (70.5)	1190 (92.3)	850 (71.4)	721 (84.8)	627 (87.0)
No	29,324 (100.0)	25,185 (85.9)	18,116 (71.9)	16,588 (91.6)	11,507 (69.4)	9917 (86.2)	8751 (88.2)
Diabetes							
At baseline	1801 (100.0)	1538 (85.4)	1153 (75.0)	1061 (92.0)	803 (75.7)	695 (86.6)	628 (90.4)
Ever	4682 (100.0)	4167 (89.0)	3022 (72.5)	2820 (93.3)	2132 (75.6)	1827 (85.7)	1605 (87.8)
Never	26,757 (100.0)	22,846 (85.4)	16,383 (71.7)	14,958 (91.3)	10,225 (68.4)	8811 (86.2)	7773 (88.2)
Major mental health illness							
At baseline	8685 (100.0)	7628 (87.8)	5391 (70.7)	4957 (91.9)	3378 (68.1)	2882 (85.3)	2497 (86.6)
Ever	17,734 (100.0)	15,678 (88.4)	11,266 (71.9)	10,488 (93.1)	7441 (70.9)	6381 (85.8)	5510 (86.4)
Never	13,705 (100.0)	11,335 (82.7)	8139 (71.8)	7290 (89.6)	4916 (67.4)	4257 (86.6)	3868 (90.9)
Alcohol use disorder							
At baseline	3095 (100.0)	2680 (86.6)	1928 (71.9)	1732 (89.8)	1099 (63.5)	918 (83.5)	764 (83.2)
Ever	7824 (100.0)	6973 (89.1)	5028 (72.1)	4618 (91.8)	3044 (65.9)	2590 (85.1)	2154 (83.2)
Never	23,615 (100.0)	20,040 (84.9)	14,377 (71.7)	13,160 (91.5)	9313 (70.8)	8047 (86.4)	7223 (89.8)
HBV coinfection							
At baseline	1002 (100.0)	806 (80.4)	531 (65.9)	492 (92.7)	377 (76.6)	308 (81.7)	266 (86.4)
Ever	1230 (100.0)	1001 (81.4)	647 (64.6)	600 (92.7)	454 (75.7)	370 (81.5)	318 (85.9)
Never	30,209 (100.0)	26,011 (86.1)	18,758 (72.1)	17,178 (91.6)	11,903 (69.3)	10,268 (86.3)	9060 (88.2)
HIV coinfection							
At baseline	1272 (100.0)	1198 (94.2)	901 (75.2)	886 (98.3)	752 (84.9)	686 (91.2)	576 (84.0)
Ever	2169 (100.0)	2058 (94.9)	1532 (74.4)	1496 (97.7)	1243 (83.1)	1118 (89.9)	935 (83.6)
Never	29,270 (100.0)	24,954 (85.3)	17,873 (71.6)	16,282 (91.1)	11,114 (68.3)	9520 (85.7)	8443 (88.7)
Decompensated cirrhosis							
At baseline	675 (100.0)	619 (91.7)	490 (79.2)	464 (94.7)	352 (75.9)	294 (83.5)	255 (86.7)
Ever	7539 (100.0)	7343 (97.4)	5907 (80.4)	5781 (97.9)	4681 (81.0)	4149 (88.6)	3626 (87.4)
Never	23,900 (100.0)	19,669 (82.3)	13,498 (68.6)	11,997 (88.9)	7676 (64.0)	6489 (84.5)	5752 (88.6)
Hepatocellular carcinoma							
At baseline	251 (100.0)	233 (92.8)	225 (96.6)	223 (99.1)	199 (89.2)	163 (81.9)	137 (84.0)
Ever	731 (100.0)	701 (95.9)	626 (89.3)	611 (97.6)	513 (84.0)	455 (88.7)	381 (83.7)
Never	30,708 (100.0)	26,312 (85.7)	18,779 (71.4)	17,167 (91.4)	11,844 (69.0)	10,183 (86.0)	8997 (88.4)

^a^ Diagnostic codes for diabetes, mental health illness, alcohol and drug consumption disorders, HIV, and liver-disease excluded.

**Table 2 viruses-16-00389-t002:** Predictors of not having received RNA testing or initiated treatment as of 2018, and changes since 2013 among individuals diagnosed with HCV since 1990 and alive as of 2013/2018, Quebec, Canada.

Characteristics ^a^	Model A: Not Having Received RNA Testing		Model B: Not Having Initiated Treatment	
	2013 ^b^ aORs (95% CI)	2018/Overall ^b^ aORs (95% CI)	2013 ^b^ aORs (95% CI)	2018/Overall ^b^ aORs (95% CI)
Sex				
Male (Ref)	--	1 (Ref)	--	1 (Ref)
Female	--	1.00 (0.93–1.07)	--	1.05 (0.99–1.12)
Birth cohort				
<1945	--	1.35 (1.16–1.57)	--	2.16 (1.89–2.46)
1945–1965	--	1.03 (0.97–1.11)	--	0.86 (0.82–0.91)
>1965 (Ref)	--	1 (Ref)	--	1 (Ref)
PWID				
No (Ref)	1 (Ref)	1 (Ref)	1 (Ref)	1 (Ref)
Yes	0.83 (0.76–0.90)	0.67 (0.61–0.74)	1.30 (1.20–1.41)	1.03 (0.96–1.11)
Immigrant status				
Non-immigrant (Ref)	--	1 (Ref)	1 (Ref)	1 (Ref)
Immigrant	--	0.76 (0.67–0.85)	0.32 (0.29–0.35)	0.63 (0.57–0.70)
Region of residence				
Rural	1.34 (1.25–1.44)	1.23 (1.14–1.33)	--	1.02 (0.96–1.08)
Urban (Ref)	1 (Ref)	1 (Ref)	--	1 (Ref)
Material deprivation index				
1 (Most privileged) (Ref)	--	1 (Ref)	--	1 (Ref)
2	--	1.21 (1.06–1.38)	--	1.04 (0.94–1.15)
3	--	1.15 (1.01–1.30)	--	1.14 (1.04–1.26)
4	--	1.19 (1.05–1.34)	--	1.20 (1.09–1.32)
5 (Most deprived)	--	1.38 (1.23–1.54)	--	1.39 (1.27–1.52)
Social deprivation index				
1 (Most privileged) (Ref)	--	1 (Ref)	1 (Ref)	1 (Ref)
2	--	0.96 (0.81–1.14)	1.20 (1.05–1.37)	1.04 (0.91–1.19)
3	--	1.04 (0.89–1.21)	1.16 (1.02–1.33)	1.05 (0.92–1.21)
4	--	1.08 (0.93–1.26)	1.40 (1.23–1.59)	1.16 (1.03–1.31)
5 (Most deprived)	--	1.13 (0.98–1.30)	1.46 (1.29–1.64)	1.22 (1.09–1.37)
Elixhauser comorbidity index (modified)				
No (Ref)	--	1 (Ref)	1 (Ref)	1 (Ref)
Yes	--	0.99 (0.84–1.16)	1.01 (0.87–1.18)	1.15 (0.99–1.34)
Diabetes				
No (Ref)	--	1 (Ref)	1 (Ref)	1 (Ref)
Yes	--	0.97 (0.83–1.14)	0.87 (0.74–1.02)	1.34 (1.16–1.55)
Major mental health diagnosis				
No (Ref)	1 (Ref)	1 (Ref)	--	1 (Ref)
Yes	0.89 (0.81–0.97)	0.82 (0.74–0.91)	--	0.96 (0.90–1.03)
Alcohol use disorder				
No (Ref)	--	1 (Ref)	--	1 (Ref)
Yes	--	1.21 (1.08–1.36)	--	1.41 (1.26–1.57)
HBV coinfection				
No (Ref)	--	1 (Ref)	--	1 (Ref)
Yes	--	1.52 (1.26–1.84)	--	0.80 (0.68–0.95)
HIV coinfection				
No (Ref)	1 (Ref)	1 (Ref)	1 (Ref)	1 (Ref)
Yes	0.30 (0.22–0.41)	0.39 (0.27–0.56)	0.93 (0.77–1.11)	0.38 (0.31–0.46)
Decompensated cirrhosis				
No (Ref)	--	1 (Ref)	--	1 (Ref)
Yes	--	0.61 (0.43–0.87)	--	1.15 (0.90–1.47)
Hepatocellular carcinoma				
No (Ref)	--	1 (Ref)	1 (Ref)	1 (Ref)
Yes	--	0.38 (0.17–0.84)	0.80 (0.41–1.56)	0.20 (0.09–0.43)

Marginal logistic regression model estimated via generalized estimating equations (GEE). Effects for 2013 shown only for characteristics with a significant (*p* < 0.05) change from 2013 to 2018. aORs, adjusted odds ratios. Model also adjusted for calendar year of HCV diagnosis. -- purposely left blank. ^a^ characteristics at the time of HCV diagnosis; ^b^ adjusted odds ratios generated from interaction terms of each variable with the care cascade calendar year (i.e., 2013 or 2018). Only interaction terms with *p* < 0.05 were included in the final model and are shown in the table above.

## Data Availability

The data that support the findings of this study are available from the Quebec Statistics Institute (ISQ, “Institut de la statistique du Québec”), but restrictions apply to the availability of these data, which were used under contract for the current study and so are not publicly available. For more information, see https://statistique.quebec.ca/recherche.

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
