# Peer review of "Ongoing Gaps in the Hepatitis C Care Cascade during the Direct-Acting Antiviral Era in a Large Retrospective Cohort in Canada: A Population-Based Study"

_viruses, 2024, doi:10.3390/v16030389_

Round 1
Reviewer 1 Report
Comments and Suggestions for Authors
Passos-Castilho and colleagues present a well written population-based modelling study using linked data, comparing the hepatitis C care cascades in Quebec pre and post the introduction of direct acting antivirals, in the general population, and priority populations of people who inject drugs and immigrant community.
As a non-statistical reviewer I cannot comment on the methodological details of the study and encourage review by a peer with statistical expertise.
The manuscript may benefit from the following considerations:
- Are prisoners considered a priority population – are there prison hepatitis C treatment programs in Quebec, were these data collected and did incarceration at time of diagnosis effect RNA test or treatment outcomes?
- Clarification in methods of hepatitis C notification requirements (clinician and lab?) and which tests notifiable (Ab positive only, or antibody positive and RNA result), and definition of HCV diagnosed (presumably HCV antibody positive)
In Results, the demographic figures in the text do not match Table S4 for mean age, PWID and some others.
- Will the study be repeated post COVID, given health care utilisation pattern may have changed post pandemic
- What is the DAA prescribing environment in Quebec, to help explain such a successful cascade – can any medical practitioner prescribe DAA or just specialists?
Author Response
Response to Reviewers
Please find below the response to each of the reviewers' comments for Manuscript viruses-2880543, entitled: "Ongoing gaps in the hepatitis C care cascade during the direct-acting antiviral era in a large retrospective cohort in Canada: a population-based study".
All changes made to the main manuscript file, or the supplementary file have been highlighted using the “track changes” feature in Microsoft Word.
Reviewer 1 - comments:
- Are prisoners considered a priority population – are there prison hepatitis C treatment programs in Quebec, were these data collected and did incarceration at time of diagnosis effect RNA test or treatment outcomes?
Response: Prisoners are indeed considered a priority population for hepatitis C in Canada. They were, however, not included as a priority population in our study as we could not identify them in the administrative database used in this study. We focused on the three groups that account for the greatest number of HCV cases in Canada including people who inject drugs (PWID), immigrants born in countries where HCV is common, and individuals born during 1945-1965.
- Clarification in methods of hepatitis C notification requirements (clinician and lab?) and which tests notifiable (Ab positive only, or antibody positive and RNA result), and definition of HCV diagnosed (presumably HCV antibody positive)
Response: There is mandatory reporting of Hepatitis C in Quebec by both laboratories and clinicians. The public health definition of confirmed hepatitis C includes any confirmed positive HCV antibody or positive HCV RNA. Those with confirmed HCV antibodies may be with or without viremia. This has been clarified in section 2.2 Care Cascade (page 3).
- In Results, the demographic figures in the text do not match Table S4 for mean age, PWID and some others.
Response: We appreciate the reviewer’s comments and the opportunity to correct these data. We have corrected Table S4 to match the most updated version of the linked dataset, which has been used for all analyses in the manuscript. Edits are tracked. We have also updated figure 2A which now matches the numbers for the birth cohorts as presented in table 1.
- Will the study be repeated post COVID, given health care utilization pattern may have changed post pandemic
Response: We agree with the reviewers that this is an important issue. Recent 2021 surveillance data from the Public Health agency of Canada have showed that by the end of 2021, the decrease in HCV diagnoses in Canada attributed to the disruption of health services during the pandemic had yet not been overcome (https://www.canada.ca/en/public-health/services/publications/diseases-conditions/hepatitis-c-canada-2021-surveillance-data-update.html). Our dataset is only up to 2018 so we unfortunately are not able to report on the impact of COVID on HCV diagnoses or health care utilization in our study.
- What is the DAA prescribing environment in Quebec, to help explain such a successful cascade – can any medical practitioner prescribe DAA or just specialists?
Response: Any medical practitioner may prescribe DAAs in Quebec and treatment became widely available without restrictions as of March 2016. This is outlined in the first paragraph on page 3 of our manuscript: “Second generation, interferon-free DAA therapy was first introduced in Canada in 2014 with unrestricted access in Quebec as of March 2016”. HCV treatments are covered by different reimbursement mechanisms but costs are covered for almost all patients; about half of the population in Quebec has publicly funded drug prescription coverage and the other half are covered by different private drug insurance plans.
Reviewer 2 Report
Comments and Suggestions for Authors
The paper by Passos-Castilho et al. presents the results of comprehensive study that addressed the possible gaps in current hepatitis C cascade of care and assessed the progress compared to pre-DAA period. The findings are important for tailoring the screening strategies and improvement of linkage to care. The methodology developed by authors can be a useful tool for assessment the existing gaps in HCV-related services in other parts of the world. I do not have any major concerns regarding this study. There are few minor comments, though.
Methods
1. Please specify the exact meaning of the first step of the cascade “HCV diagnosed”. Is it HCV Ab positive, considering that HCV RNA data are given as steps 2 (RNA tested) and 3 (RNA positive).
2. Why SVR was defined as a negative RNA test at least 10 weeks after the end of treatment, while the standard SVR definition is 12 weeks?
Results
Figure 2A. Colors in diagram and the flow chart should be the same. In current version, for example, dark green indicate 1945-1965 in diagram, but <1945 in flow chart below the diagram.
Discussion
Although real-world SVR rates in post-DAA era are reported to be >95%, SVR rates observed in this study in majority of cohorts were below 90%, and varied substantially between cohorts. Authors should discuss the possible explanations for this.
Author Response
Response to Reviewers
Please find below the response to each of the reviewers' comments for Manuscript viruses-2880543, entitled: "Ongoing gaps in the hepatitis C care cascade during the direct-acting antiviral era in a large retrospective cohort in Canada: a population-based study".
All changes made to the main manuscript file, or the supplementary file have been highlighted using the “track changes” feature in Microsoft Word.
Reviewer 2 - comments:
- Methods: Please specify the exact meaning of the first step of the cascade “HCV diagnosed”. Is it HCV Ab positive, considering that HCV RNA data are given as steps 2 (RNA tested) and 3 (RNA positive).
Response: The public health definition of confirmed hepatitis C includes any confirmed positive HCV antibody or a positive HCV RNA (a very small proportion of all cases). Those with confirmed HCV antibodies may be with or without viremia. This has been clarified in section 2.2 Care Cascade (page 3).
- Methods: Why SVR was defined as a negative RNA test at least 10 weeks after the end of treatment, while the standard SVR definition is 12 weeks?
Response: The standard definition is indeed 12 weeks. We use 10 weeks in this study to align our methods, with the BC group (co-author Naveed Janjua) who has published on the BC care cascade previously. In their cohort (which has complete treatment data), they looked at the distribution of RNA tests post treatment and used a cut off of week 10 so that they would not lose patients who had their test done just before week 12.
- Results: Figure 2A. Colors in diagram and the flow chart should be the same. In current version, for example, dark green indicate 1945-1965 in diagram, but <1945 in flow chart below the diagram.
Response: We appreciate the comment and the opportunity to correct the figure 2A. The numbers for each birth cohort and the color legend now matches the numbers for the birth cohorts as presented in table 1.
- Discussion: Although real-world SVR rates in post-DAA era are reported to be >95%, SVR rates observed in this study in majority of cohorts were below 90%, and varied substantially between cohorts. Authors should discuss the possible explanations for this.
Response: Thanks very much for this comment. The data presented in table 1 and figures 1 & 2 are the cumulative HCV care cascade from 1990-2018 and thus the SVR presented includes SVR from both the pre and post-DAA periods. The small differences across groups is likely due to specific demographic characteristics where compliance with medication may be lower such as for PWID. The lower SVR in younger birth cohorts, males and those with major mental illness is likely because many are also PWIDS.